# *SlTDC1* Overexpression Promoted Photosynthesis in Tomato under Chilling Stress by Improving CO_2_ Assimilation and Alleviating Photoinhibition

**DOI:** 10.3390/ijms241311042

**Published:** 2023-07-03

**Authors:** Xutao Liu, Yanan Wang, Yiqing Feng, Xiaowei Zhang, Huangai Bi, Xizhen Ai

**Affiliations:** State Key Laboratory of Crop Biology, Key Laboratory of Crop Biology and Genetic Improvement of Horticultural Crops in Huanghuai Region, Collaborative Innovation Center of Fruit & Vegetable Quality and Efficient Production in Shandong, College of Horticulture Science and Engineering, Shandong Agricultural University, Tai’an 271018, China; liuxutaosd@163.com (X.L.); 18864805562@163.com (Y.W.); 17863805447@163.com (Y.F.); zxw@sdau.edu.cn (X.Z.)

**Keywords:** *SlTDC1*, tomato, photosynthesis, photoinhibition, melatonin synthesis, chilling stress

## Abstract

Chilling causes a significant decline in photosynthesis in tomato plants. Tomato tryptophan decarboxylase gene 1 (*SlTDC1*) is the first rate-limiting gene for melatonin (MT) biosynthesis and is involved in the regulation of photosynthesis under various abiotic stresses. However, it is not clear whether *SlTDC1* participates in the photosynthesis of tomato under chilling stress. Here, we obtained *SlTDC1* overexpression transgenic tomato seedlings, which showed higher *SlTDC1* mRNA abundance and MT content compared with the wild type (WT). The results showed that the overexpression of *SlTDC1* obviously alleviated the chilling damage to seedlings in terms of the lower electrolyte leakage rate and hydrogen peroxide content, compared with the WT after 2 d of chilling stress. Moreover, the overexpression of *SlTDC1* notably increased photosynthesis under chilling stress, which was related to the higher chlorophyll content, normal chloroplast structure, and higher mRNA abundance and protein level of Rubisco and RCA, as well as the higher carbon metabolic capacity, compared to the WT. In addition, we found that *SlTDC1*-overexpressing seedlings showed higher W_k_ (damage degree of OEC on the PSII donor side), φ_Eo_ (quantum yield for electron transport in the PSII reaction center), and PI_ABS_ (photosynthetic performance index) than WT seedlings after low-temperature stress, implying that the overexpression of *SlTDC1* decreased the damage to the reaction center and donor-side and receptor-side electron transport of PSII and promoted PSI activity, as well as energy absorption and distribution, to relieve the photoinhibition induced by chilling stress. Our results support the notion that *SlTDC1* plays a vital role in the regulation of photosynthesis under chilling stress.

## 1. Introduction

Low-temperature stress is a serious threat to warm-season plants at various developmental stages and significantly limits the growth and yield of these plants. Usually, low-temperature stress includes freezing and chilling (above freezing) conditions. A large number of studies have proven that cold stress causes plant membrane damage in terms of an increase in the relative electrolyte leakage (EL) and leads to the accumulation of reactive oxygen species (ROS), thus disrupting the redox balance; injuring cell membranes, nucleic acids, proteins, and lipids; and impairing many physiological processes [1,2,3]. Photosynthesis is one of the metabolic processes of plants that is most sensitive to chilling stress. Chilling stress significantly decreases the expression and activities of photosynthetic enzymes [4], destroys the chloroplast structure, and reduces chlorophyll content. PSII is a major protein-pigment complex involved in photosynthetic electron transport in chloroplast thylakoid membranes and is responsible for light-driven water oxidation and plastoquinone reduction, the inhibition of which is one of the most prominent features during low-temperature stress [5]. Similar to PSII, chilling stress also causes a decline in PSI in terms of the decrease in ΔI/I_0_ [6].

A number of studies have proven that melatonin (MT), as a plant hormone, participates in the regulation of abiotic stress tolerance, including drought, salt, and high temperature tolerance [7,8,9]. Melatonin acts on plant photosynthesis by regulating many aspects of the photosystem, electron transport chain, and Calvin–Benson cycle under abiotic stresses [4,10,11,12]. For warm-climate plants, chilling stress often occurs and causes damage to growth and yield, and MT was first reported to participate in the regulation of chilling stress in carrot suspension cells [13]. Subsequently, the application of MT was proven to increase the chilling tolerance of plants [14,15,16]. According to the study of Ding [17], melatonin can not only reduce the level and content of reactive oxygen species, improve the activity of antioxidant enzymes, and promote the expression of cold response genes but can also increase photosynthetic carbon assimilation to improve the cold tolerance of the tomato. Moreover, the application of exogenous MT improved photosynthetic characteristics by increasing chlorophyll content and upregulating the expression of PSII reaction central-related protein to finally promote chilling tolerance [18].

The synthesis of MT in plants is catalyzed by tryptophan decarboxylase (TDC), serotonin N-acetyltransferase (SNAT), and N-acetyl-5-serotonin-methyltransferase (ASMT), with tryptophan as the precursor substance, and the encoding genes for these enzymes have been cloned and used to change the endogenous MT content via genetic transformation techniques to study the role of MT in the regulation of abiotic stress [19,20]. For example, the overexpression of *ASMT* alleviated heat stress by ameliorating heat-induced photoinhibition, decreasing EL, and upregulating the expression of HSP proteins in tomato plants [21]. Meanwhile, the overexpression of *SNAT* could also increase the endogenous MT content to scavenge excessive ROS and then promote drought tolerance. In addition, the heterologous expression of *SNAcT* from humans could upregulate the expression of *CBFs/DREBs*, *COR15a*, *CAMTA1*, *ZAT10*, and *ZAT12* and then activate downstream physiological metabolic pathways to promote the chilling tolerance of plants [22]. *TDC* was the first gene identified during the synthesis of MT which showed a positive relationship with endogenous MT content [23,24]. The aging process of rice leaves with overexpression of *TDC* gene was slower than that of the wild type, and the aging process of rice leaves was accelerated after *TDC* gene expression was inhibited, indicating that the *TDC* gene affected the synthesis of melatonin content, thus affecting the aging process [24]. Furthermore, the developmental roles of *TDC* and serotonin were revealed through the observation and analysis of overexpressors, in which early fruit ripening was observed [25]. However, the regulatory effect of *TDC* on photosynthesis under chilling stress in plants is still ambiguous.

The tomato (*Solanum lycopersicum* L.) is one of the worldwide vegetables and is often cultivated in facilities in Northern China. This plant often encounters chilling stress during the winter and spring seasons, causing obvious decreases in photosynthesis and leading to a decline in growth and yield. In recent years, significant progress has been made in studying the effects of melatonin on plant tolerance to low-temperature stress, but it is still unclear how *SlTDC1* regulates photosynthesis under chilling stress and whether *SlTDC1* could improve endogenous melatonin to improve the chilling tolerance of the tomato. Therefore, this study obtained *SlTDC1*-overexpressing transgenic seedlings; studied the difference in chilling tolerance between *SlTDC1*-overexpressing transgenic seedlings and wild-type seedlings; and elucidated the effect of *SlTDC1* on chloroplast structure, electron transport efficiency, and carbon assimilation in the tomato under chilling stress.

## 2. Results and Discussion

### 2.1. Electrolyte Leakage Rate and H_2_O_2_ Content

In our study, we obtained nine *SlTDC1* overexpression transgenic lines via Agrobacterium-mediated transformation, and as shown in Appendix A, the relative expression of *SlTDC1* in the T2 generation was still significantly higher than that in the WT. OE10 and OE14 transgenic lines were selected for subsequent experiments. First, we detected the MT contents of OE10 and OE14. The data showed that the endogenous MT content of OE10 and OE14 seedlings increased by 3.29- and 2.8-fold, respectively, compared to that of WT seedlings, implying that the overexpression of *SlTDC1* indeed promoted the synthesis of endogenous MT (Figure 1). To verify the response of *SlTDC1* overexpression to chilling stress, we further measured the change in EL and H_2_O_2_ content in both the WT and transgenic seedlings. As shown in Figure 2, chilling stress induced an increase in the EL and H_2_O_2_ contents, whereas lower EL and H_2_O_2_ contents were obtained in OE14 and OE10 seedlings; thus, WT seedlings showed more serious damage after 2 days of chilling stress.

### 2.2. Photosynthetic Parameter

Photosynthesis is sensitive to chilling conditions, and we found that the Pn, A_sat_, V_cmax_, and J_max_ of both WT and *SlTDC1*-overexpressing seedlings significantly declined after one day of chilling stress, which gradually decreased with prolonged lower-temperature treatment time (Figure 3a and Figure 4a–c). However, at the end of the stress, the Pn, A_sat_, V_cmax_, and J_max_ in transgenic seedlings were still higher than those in the WT plants. For instance, the Pn of OE14 and OE10 seedlings was 43.84% and 41.23% higher, respectively, than that of WT seedlings. In addition, we found similar changes in Gs and E and opposite changes in Ci to Pn in tomato seedlings under low-temperature stress, implying that the photosynthesis decrease in tomato seedlings was caused mainly by the nonstomatal limitation (Figure 3b–d). As shown in the figure, Pn, Gs, and E decreased significantly and Ci increased significantly in wild-type and all lines of tomato seedlings subjected to low-temperature stress, but the Pn, Gs, and E of tomato seedlings overexpressing *SlTDC1* were higher than those with inhibited expression, the differences were significant, and the trend of Ci was the opposite. These results indicated that *SlTDC1*-overexpressing tomato seedlings had a high endogenous melatonin content, which alleviated the damage of low-temperature stress on the photosynthetic apparatus of tomato leaves and improved the photosynthetic performance.

### 2.3. Gene Expression and Photosynthetic Enzyme Activity

Rubisco is the key enzyme of the Calvin cycle. Here, we further detected the change in the mRNA abundance and activities of Rubisco and RCA in both WT and *SlTDC1*-overexpressing seedlings under chilling stress. Similar to our previous results, chilling notably decreased the relative expression and activity of Rubisco in tomato seedlings. Importantly, *SlTDC1*-overexpressing seedlings with higher MT contents showed higher mRNA abundance and activity of Rubisco not only before chilling stress but also during chilling stress (Figure 5a,b,d). We also investigated the change in RCA, which mainly activates Rubisco, in WT and *SlTDC1*-OE plants and found that the transcript level and activity of RCA were obviously higher in *SlTDC1*-OE plants than in WT plants both before and after chilling stress (Figure 5c,e).

### 2.4. Chlorophyll Content and Chloroplast Ultrastructure

As shown in Figure 5a,b, *SlTDC1* overexpression significantly increased the chlorophyll a and b contents, which started to decrease after 1 day of chilling stress. Compared to WT seedlings, *SlTDC1*-OE seedlings displayed higher chlorophyll contents after 1 and 2 days of treatment at 8/5 °C, implying that *SlTDC1* overexpression significantly alleviated the chlorophyll degradation induced by low-temperature stress. Meanwhile, we observed the change in chloroplast ultrastructure in *SlTDC1*-OE seedlings after chilling stress and found that *SlTDC1* overexpression maintained the relatively intact chloroplast structure, as evidenced by a long elliptic structure close to the cell wall, lower starch grains and osmophore, and distinct cytomembrane and substrate lamellar after 2 days of chilling stress (Figure 6i–iv).

### 2.5. Carbon Metabolism

Carbon metabolism is an important physiological process for carbon conversion. In accordance with the change in starch grains in chloroplasts, the data showed that the WT seedlings accumulated more starch content than the *SlTDC1*-OE seedlings did after chilling stress. In contrast to the result with the starch content, chilling induced more sucrose, glucose, fructose, total sugar content, and A_sat_ in the leaves of *SlTDC1*-OE seedlings than in those of the WT seedlings (Table 1). To confirm that *SlTDC1* indeed positively regulated carbon metabolism, the gene expression and activities of sucrose synthase (SS) and sucrose phosphate synthase (SPS) were further measured. As shown in Figure 7, the mRNA abundance and activities of SS and SPS in *SlTDC1*-OE plants were dramatically higher than those in WT plants. Collectively, these results suggest that *SlTDC1*-OE seedlings with higher MT content indeed promoted carbon accumulation and transport under chilling stress.

### 2.6. Photoinhibition

Photoinhibition induced by abiotic stresses includes PSI and PSII photoinhibition, which further lead to a decrease in photosynthetic efficiency. Here, we found a significant decline in PSI and PSII activities in all seedlings after 2 days of chilling stress, while *SlTDC1*-overexpressing tomato seedlings showed higher PSI and PSII activities than WT seedlings in terms of higher Fv/Fm, ΦPSII, and ΔI/I_0_ in *SlTDC1*-OE 10 and 14 seedlings (Figure 8a,b). With prolonged low-temperature treatment, NPQ increased, qP and ETR decreased, NPQ was lower, and qP and ETR were higher in *SlTDC1*-OE seedlings than in WT seedlings at the end of treatment (Figure 8c–e). These results implied that *SlTDC1* overexpression obviously alleviated chilling-induced photoinhibition and improved electron transport under chilling stress.

### 2.7. Electron Transport and Energy Distribution

To further confirm the effect of *SlTDC1* overexpression on PSII under chilling stress, an O-J-I-P curve of tomato seedlings was observed. The data showed that the morphology of the O-J-I-P curve of tomato leaves changed significantly in terms of a decrease in the I and P points after 2 d of low-temperature stress. Then, we standardized the O-J phase and found that W_k_ increased notably, while the *SlTDC1*-OE seedlings showed lower W_k_ than the WT seedlings, implying that *SlTDC1* overexpression decreased the damage to OEC induced by chilling stress (Figure 9a–c,e). In parallel, we also detected injury to the PSII reaction center and PSII acceptor side in both WT and transgenic seedlings during chilling stress. Chilling stress also caused a decline in φ_Eo_ and PI_ABS_. However, in the *SlTDC1*-OE seedlings, the φ_Eo_ and PI_ABS_ were obviously 14.67% and 47.12% higher than those in the WT ones after 2 d of chilling stress (Figure 9d,f).

To investigate how *SlTDC1* overexpression affects light energy absorption and distribution, we measured and obtained a phenomenological pipeline model of energy fluxes which showed that chilling stress significantly reduced ABS/CS_m_, TR/CS_m_, D/CS_m_, and E/CS_m_. The *SlTDC1* overexpression in ABS/CS_m_, TR/CS_m_, D/CS_m_, and E/CS_m_ under chilling stress was higher than that in the WT under chilling stress. In addition, chilling stress increased the density of inactive reaction centers, as reflected by the increase in the number of closed circles in the tomato seedlings, and the number of hollow circles in *SlTDC1* overexpression was greater than that in the WT (Figure 10), indicating that the density of active reaction centers in *SlTDC1* overexpression was higher under chilling stress.

### 2.8. Photosynthesis-Related Protein Level

To further confirm the role of *SlTDC1* overexpression in the PSI and PSII reaction centers under chilling stress, protein levels such as RCA, rbcL, Psbs, and D1 were determined. The immunoblot analysis revealed that obvious decreases in RCA, rbcL, Psbs, and D1 protein levels were observed under chilling stress compared with the levels under normal conditions, and *SlTDC1* overexpression delayed the declines in the levels of these proteins, as evidenced by reductions in RCA, rbcL, Psbs, and D1 occurring under chilling stress (Figure 11).

## 3. Materials and Methods

### 3.1. Plant Material and Growth Conditions

Tomato plants (*Solanum lycopersicum* ‘cv. Ailsa Craig’) and *SlTDC1* overexpression lines (generated previously in our laboratory) were used as experimental materials to study the chilling tolerance of tomatoes induced by *SlTDC1*. The wild-type (WT) and T2-generation *SlTDC1* seeds were sterilized with distilled water at 55 °C for 10 min, immersed in water for 10 h, and then incubated on wet filter paper (28 ± 1 °C) for 48 h to accelerate germination. After germination, all the seeds were sown in a 10 cm diameter pot and grown in a greenhouse. The conditions were as follows: the average temperature of day/night was approximately 25 °C/18 °C, the average light flux density (PFD) was approximately 800 μmol·m^−2^·s^−1^, and the photoperiod was 14/10 h.

For the chilling-tolerance experiment, the WT and transgenic tomato seedlings were both transferred to chilling conditions (8 °C/5 °C) at the four-leaf stage, simulated by a climate chamber, and the tissues were sampled or measured for parameters after chilling stress for 0 d, 1 d, and 2 d.

### 3.2. Plant Transformation

The full-length *SlTDC1* cDNA and 401 bp *SlTDC1* loop were inserted into a modified pBWA(V)HS vector. Then, the recombinant vectors were introduced into A. tumefaciens strain LBA4404. AC tomato was used for transformation while following a previously published protocol [26]. Transgenic plants were validated by PCR and qPCR. *SlTDC1*-OE10 and *SlTDC1*-OE14 were purified and used to conduct the experiment in this paper.

### 3.3. Determination of MT Content

The 0-day-, 1-day-, and 2-day-old leaves of chilling-treated tomato plants were collected and immediately frozen in liquid nitrogen. Frozen leaves were pulverized, and 0.1 g of powder was extracted from each sample for MT, as described previously. Then, 10 μL of each extract solution was analyzed on an HPLC–MS system (Thermo Fisher Scientific, TSQ Quantum Access, Waltham, MA, USA), and the testing conditions were set according to Liu [27].

### 3.4. Determination of EL, H_2_O_2_, and Chlorophyll Content

The electrolyte leakage rate (EL) and pigment content were measured according to previously described methods [28]. The hydrogen peroxide (H_2_O_2_) content was detected according to the instructions of the H_2_O_2_ kit (A064-1-1, Nanjing Jiancheng Bioengineering Institute, Nanjing, China).

### 3.5. Determination of Chloroplast Ultrastructure

Square 1–2 mm pieces on both sides of the main vein of the target leaves were cut with a sterilized blade and fixed in 2.5% glutaraldehyde for more than 4 h. The subsequent processing was carried out according to the method of Liu [27], and the observation of samples was conducted with a JEM-1200 EX transmission electron microscope (Tokyo, Japan).

### 3.6. Determination of Gas-Exchange Parameters and Photosynthetic Enzyme Activity

The gas-exchange parameters, such as Pn(Net photosynthetic rate), Gs(Stomatal conductance), E(Transpiration rate), and Ci(Intercellular CO_2_ concentration), were measured by a PP-Systems CIRAS-3 photosynthesis apparatus (Hitchin, Hertfordshire, UK). During the measurement time, the PFD was set to 600 μmol·m^−2^·s^−1^, the CO_2_ concentration was set to 400 μL·L^−1^, and the leaf temperature was set to 25 ± 1 °C. Meanwhile, we measured the response curves of the photosynthetic rate (Pn) to PFD or to CO_2_ concentration, and then we calculated the maximum regeneration rate of ribulose-1,5-bisphosphate carboxylase/oxygenase (J_max_), the light-saturated photosynthetic rate (A_sat_), and the maximum rate of carboxylation of Rubisco (V_cmax_), according to the method of Bernacchi [29] and McMurtrie and Wang [30].

The activities of Rubisco and Rubisco activase (RCA) were determined using kits (RUBPS-2B-Y and MM-0602O1) from Suzhou Keming Biotechnology Co., Ltd., Suzhou, China.

### 3.7. Determination of Chlorophyll Fluorescence Parameters and OJIP Curve

The photochemical efficiency (Fv/Fm) and maximum photochemical efficiency of PSII in darkness (ΦPSII) were measured and visualized with a variable chlorophyll fluorescence imaging system (Imaging PAM, Walz) according to the method of Tian [31]. NPQ, qP, and ETR were detected using the FMS-2 pulse-modulated fluorometer (Hansatech, King’s Lynn, Norfolk, UK), as described previously [28]. The chlorophyll fluorescence imaging and analysis were visualized with a variable chlorophyll fluorescence imaging system. O-J-I-P curves, W_k_ (damage degree of OEC on the PSII donor side), φEo (quantum yield for electron transport in the PSII reaction center), PI_ABS_ (photosynthetic performance index), ABS/CS_m_ (photon flux absorbed by pigment per CS antenna), TR/CS_m_ (captured energy flux per CS), ET/CS_m_ (electron transport flux per CS), D/CS_m_ (non-photochemical quenching), and 820 nm transmission were measured using an integral multifunctional plant efficiency analyzer (M-PEA, Hansatech, King’s Lynn, Norfolk, UK) and calculated according to previous methods [32,33,34].

### 3.8. Determination of Sugar Content and Related Enzyme Activities

The contents of soluble sugar, sucrose, and starch were determined by anthrone colorimetry, and the contents of fructose and glucose were determined by kits (GT-2-Y, PT-2-Y) purchased from Nanjing Jiancheng Bioengineering Institute. The activities of sucrose synthase (SS) and sucrose phosphate synthase (SPS) were measured with kits (SISI-1-Y and SPS-2-Y) produced by Suzhou Keming Biotechnology Co., Ltd. (Suzhou, China).

### 3.9. Determination of Gene Expression and Protein Level

Total RNA was extracted from tomato leaves, using a TRIzol kit from Tiangen Biochemical Technology Co., Ltd. (Beijing, China), and reverse transcribed with the HiScript R III RT SuperMix for qPCR (+gDNA wiper) (Vazyme, Nanjing, China). Assays were performed using a real-time PCR kit from Kangwei Century Biotechnology (Taizhou, China). The primers listed in Table 2 were designed using Primer 6.0 and synthesized by Ruboshengke Biotechnology Co., Ltd. (Beijing, China). Roche real-time PCR (LightCycler 480II, Penzberg, Germany) was used for the determination of gene expression. The expression level for each sample was calculated as 2^−ΔΔCt^, where Ct represents the cycle number when the fluorescence signal in each reaction reaches the threshold.

Total protein was extracted from tomato leaves and then loaded and separated on 8% (*w*/*w*) SDS–PAGE gels. The denatured protein complex was transferred to a PVDF membrane by wet transfer and then blocked with 5% (*w*/*w*) skim milk powder. After blocking, the cells were incubated with primary antibody for 1.5 h and then incubated with secondary antibody for 1 h. Finally, the color reaction was performed by the chemiluminescence method. Detection was performed with ChemiDoc™ XRS (Shanghai, China), a Bio-Rad gel imager. During the immunoblot analysis, antibodies specific to the RCA, RbcL, D1, and Psbs proteins (AT2G39730, ATCG00490, ATCG00020, and AT1G44575; PhytoAB company, San Francisco, CA, USA) were used to detect RCA, RbcL, PsaD, D1, and Psbs, followed by incubation with horseradish-peroxidase-conjugated anti-rabbit IgG antibody (ComWin Biotech Co., Ltd., Beijing, China).

### 3.10. Statistical Analysis

Microsoft Excel was used for data analysis, SigmaPlot10.0 software was used for plotting, DPS software was used for one-way analysis of variance, and Duncan’s test was used for multiple comparisons of significance (*p* < 0.05). Data in this paper are shown as the mean ± standard deviation (SD), and the experiments were designed with at least three replicates.

## 4. Discussion

Numerous studies have shown that MT can significantly regulate the chilling tolerance of plants [17,22]. Here, we also found that *SlTDC1*-overexpression seedlings with higher endogenous melatonin content (Figure 1) showed slight chilling damage in terms of lower EL and H_2_O_2_ content compared with WT seedlings (Figure 2); these results were similar to the results of the application of MT on cucumber seedlings under chilling stress [34] and implied that *SlTDC1* was involved in the chilling tolerance induced by MT.

Photosynthesis contributed approximately 90% of the dry matter, which obviously declined under chilling stress. Studies have illustrated that the decrease in the Pn of plants induced by chilling stress was mainly caused by nonstomatal limitation [26]. Here, we found that both the Pn and Gs of tomato seedlings decreased; however, Ci increased notably under chilling stress, implying that nonstomatal factors led to a decrease in Pn, but the Pn and Gs in *SlTDC1*-overexpressing plants were significantly higher than those in the WT plants under chilling stress (Figure 3). Surprisingly, we also found that *SlTDC1*-overexpressing seedlings exhibited a relatively higher V_cmax_ and J_max_ after 2 d of chilling stress than WT seedlings, and we speculated that this change might be related to the increase in the activities, mRNA, and protein levels of Rubisco and RCA in OE-*SlTDC1* tomato seedlings under chilling stress (Figure 5), which are the rate-limiting enzymes of photosynthetic CO_2_ assimilation and are usually first injured by chilling stress compared with other photosynthetic parameters [4]. Chloroplasts are the sites where plants perform photosynthesis, and the destruction of chloroplasts can combine chlorophyllase and chlorophyll and then accelerate the degradation of chlorophyll [35]. In this paper, the data showed that chilling stress led to the deformation of the chloroplast structure, loose grana lamellae, and the accumulation of starch grain, as well as lower chlorophyll content (Figure 6). Previous studies have proven that the application of exogenous MT can maintain the chloroplast structure and increase the chlorophyll content to increase the photosynthetic capacity [36,37]. Similarly, we found that *SlTDC1*-overexpression seedlings showed slight chloroplast structural damage, fewer osmophilic particles, and higher chlorophyll contents than WT seedlings under chilling stress, and this may be another important reason for the higher Pn in OE-*SlTDC1* seedlings after chilling stress. Carbon metabolism is an important physiological process of carbon conversion, and we found that chilling stress increased the carbohydrate content in tomato leaves, while *SlTDC1* overexpression further enhanced the activities and gene expression of SPS and SS, as well as sucrose, glucose, fructose, and A_sat_; however, it decreased the starch content compared to the WT, which was in accordance with the lower starch grain in OE-*SlTDC1* seedlings (Table 1 and Figure 7). Earlier studies have reported that sucrose synthesis capacity limits the maximal rates of photosynthesis by inhibiting the rate at which inorganic phosphate is recycled to support carbon fixation and electron transport in the chloroplast [38,39]. Thus, we speculated that *SlTDC1* overexpression maintained the balance between triose-phosphate transport and Pi translocation and then finally promoted the A_sat_ and carbohydrate contents of seedlings under low-temperature stress.

Photosynthetic electron transport is critical for maintaining optimal photosynthetic rates and ensuring effective energy flow for plant growth, development, and stress response [40]. Abiotic-stress-induced photoinhibition includes PSI and PSII photoinhibition, which further leads to decreased photosynthetic efficiency. Fv/Fm and ΔI/I_0_ reflected the inhibition of PSII and PSI [41]. Here, we found that chilling induced a decline in Fv/Fm and ΔI/I_0_, which were higher in *SlTDC1*-overexpressing tomato seedlings than in WT seedlings, implying that the overexpression of *SlTDC1* alleviated the photoinhibition of PSII and PSI (Figure 8). Moreover, OE-*SlTDC1* seedlings showed higher ΦPSII, qP, and ETR and lower NPQ than WT seedlings, and this was related to the higher PSII reaction-center openness, carbon assimilation capacity, and electron transport rate. The JIP test can be used to analyze the effect of environmental factors on the photosynthetic apparatus, particularly on PSII function [42]. Among the many parameters of the O-J-I-P curve, PI_ABS_, W_k_, and φ_Eo_ reflect the reaction center of PSII and its donor and acceptor sides, respectively [43]. In our previous studies, we proved that the application of MT could alleviate the PSII inhibition induced by chilling stress by decreasing the damage to the reaction center of PSII and its donor and acceptor sides, as well as reducing the degradation of PSI- and PSII-related proteins. In accordance with these results, we found that OE-*SlTDC1* seedlings also showed lower W_k_ and higher PI_ABS_, φ_Eo_, and levels of Psbs and D1 protein than WT seedlings after exposure to chilling stress, implying that *SlTDC1* participated in the regulation of MT on electron transport by decreasing damage to OEC and increasing the capacity of QA for electron transport and the related protein level. The light absorption of leaves is closely related to the chlorophyll content of leaves [44]. Tomato leaves with *SlTDC1* overexpression had higher chlorophyll content than WT leaves under chilling stress (Figure 6), so ABS/CS_m_ had higher absorbance (Figure 10). Consistent with light absorption, *SlTDC1* overexpression resulted in increased light capture compared to TR/CS_m_ in WT plants. Chilling stress decreased ET/CS_m_ in WT and *SlTDC1*-overexpressing plants, and this may be due to inactivation of response centers (Figure 10). The number of active reaction centers on the cross-section of PSII is represented by hollow black circles in the pipeline blade model of the energy flux. Ahammed et al. found that heat stress reduced the density of active reaction centers (RC/CS_m_), thus negatively affecting the photosynthetic electron transport efficiency of tomato leaves [19]. This is similar to our results. It can be seen from Figure 10 that, under chilling stress, the WT plants have significantly more solid black circles and fewer active centers than the *SlTDC1*-overexpression plants, thus reducing the photosynthetic electron transport efficiency of tomato leaves.

In conclusion, we identified the role played by *SlTDC1* in tomato photosynthesis under chilling stress. Tomato tryptophan decarboxylase gene 1 overexpression promoted the photosynthetic capacity of the tomato by maintaining the normal chloroplast structure, promoting photosynthetic carbon assimilation and carbon metabolism, and alleviating the photoinhibition of PSII and PSI in tomato seedlings under chilling stress.

## Figures and Tables

**Figure 1 ijms-24-11042-f001:**
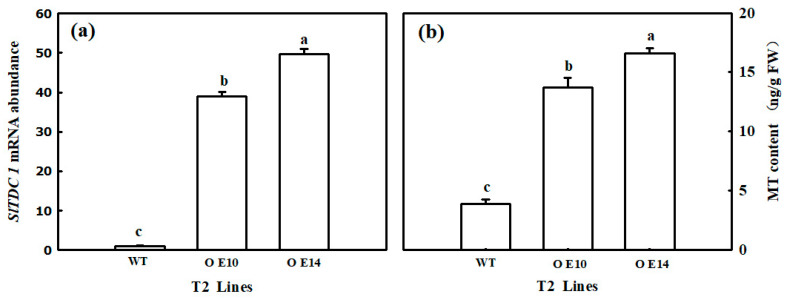
The mRNA abundance (**a**) and endogenous melatonin content (**b**) of T2-generation SITDC1-overexpression tomato plants. All values shown are the mean ± SD (*n* = 3). Lowercase letters a–c indicate that mean values are significantly different among samples (*p* < 0.05).

**Figure 2 ijms-24-11042-f002:**
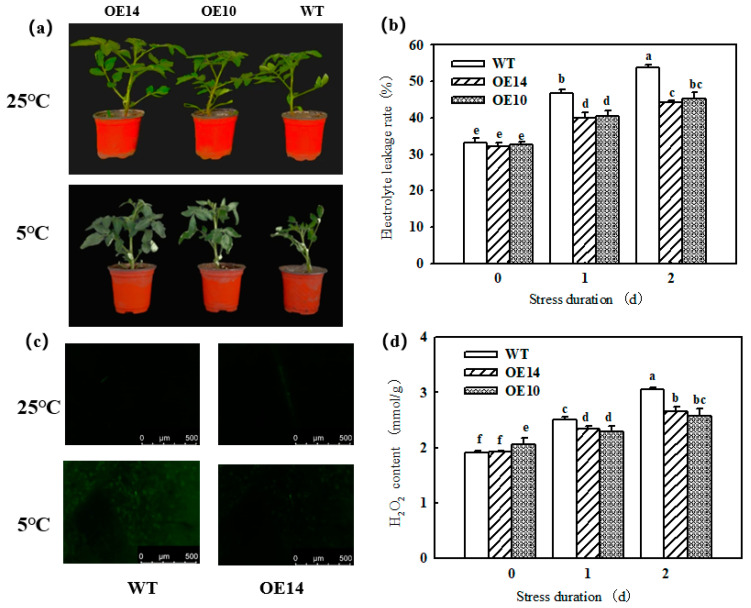
The effect of *SlTDC1* overexpression on the chilling tolerance of tomato seedlings. (**a**) The phenotype of tomato seedlings after 2 d of chilling stress. (**b**) Electrolyte leakage rate. (**c**) H_2_O_2_ inverted fluorescence microscope imaging. (**d**) H_2_O_2_ content. The four-leaf-stage tomato seedlings were exposed to 8 °C/5 °C for 2 days and sampled every day. All values shown are the mean ± SD (*n* = 3). Lowercase letters a–f indicate that mean values are significantly different among samples (*p* < 0.05).

**Figure 3 ijms-24-11042-f003:**
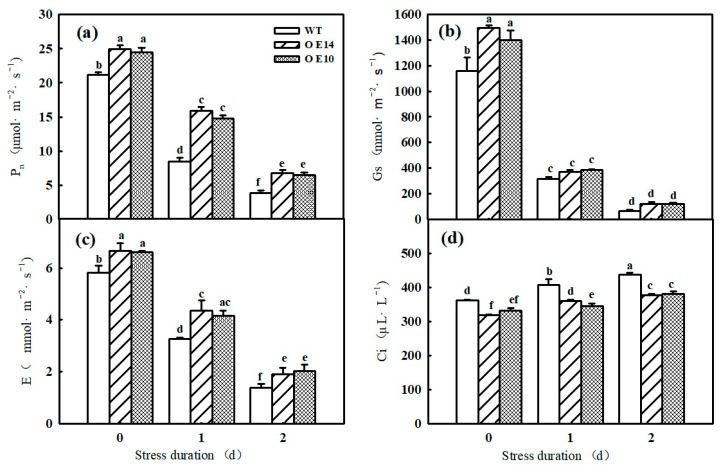
The effect of *SlTDC1* overexpression on photosynthesis in tomato seedlings under chilling stress: (**a**) P_n_, (**b**) Gs, (**c**) E, and (**d**) Ci. The four-leaf-stage tomato seedlings were exposed to 8 °C/5 °C for 2 days and measured every day. All values shown are the mean ± SD (*n* = 3). Lowercase letters a–f indicate that mean values are significantly different among samples (*p* < 0.05).

**Figure 4 ijms-24-11042-f004:**
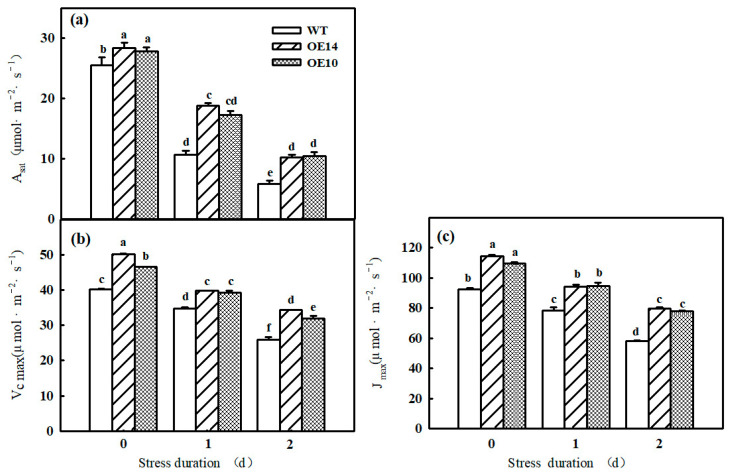
Effects of *SlTDC1* overexpression on A_sat_, V_cmax_, and J_max_ of tomato seedlings under chilling stress: (**a**) A_sat_, (**b**) V_cmax_, and (**c**) J_max_. The four-leaf-stage tomato seedlings were exposed to 8 °C/5 °C for 2 days and measured every day. All values shown are the mean ± SD (*n* = 3). Lowercase letters a–f indicate that mean values are significantly different among samples (*p* < 0.05).

**Figure 5 ijms-24-11042-f005:**
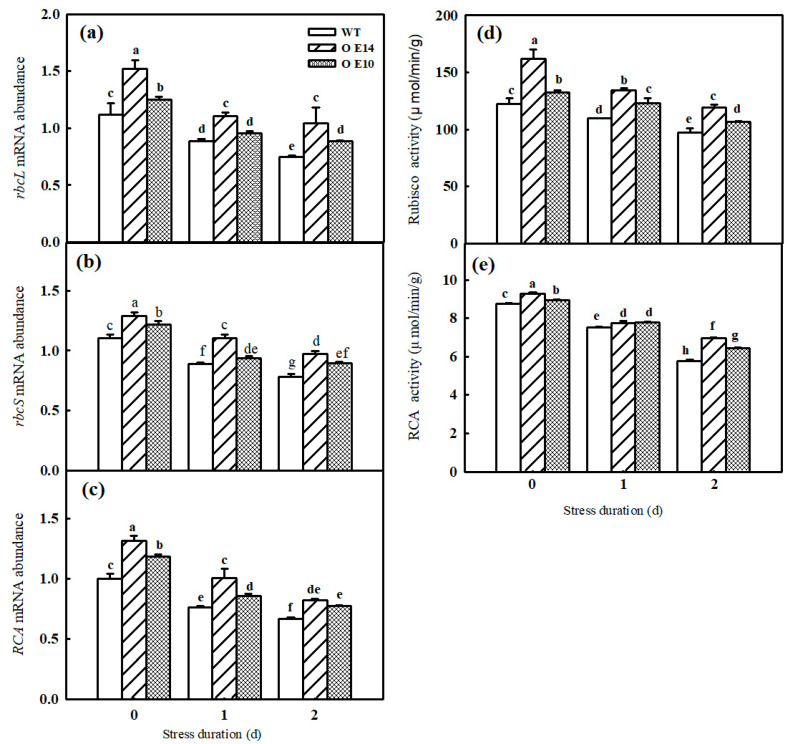
The effect of *SlTDC1* overexpression on the activities and relative mRNA expressions of enzymes in tomato seedlings under chilling stress. (**a**–**c**) *RbcL,RbcS* and *RCA* mRNA abundance. (**d**,**e**) Activities of Rubisco and RCA. The four-leaf-stage tomato seedlings were exposed to 8 °C/5 °C for 2 days and sampled every day. All values shown are the mean ± SD (*n* = 3). Lowercase letters a–h indicate that mean values are significantly different among samples (*p* < 0.05).

**Figure 6 ijms-24-11042-f006:**
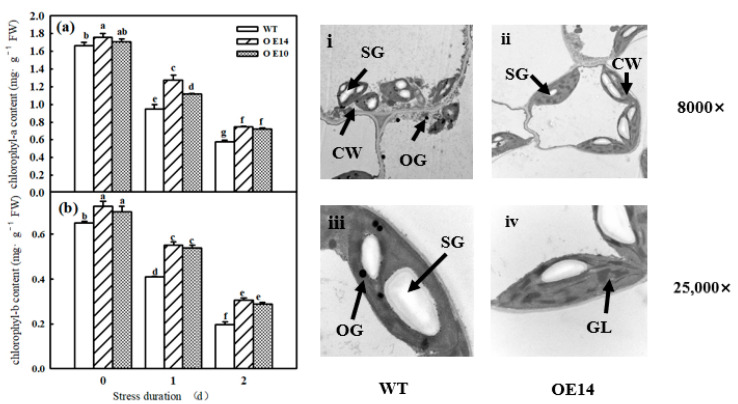
The effect of *SlTDC1* overexpression on chloroplast structure and chlorophyll in tomato seedlings under chilling stress: (**a**,**b**) chlorophyll content and (**i**–**iv**) chloroplast structure. SG, starch grain; OG, osmiophilic particles; GL, grana lamella; CW, chloroplast membrane. The four-leaf-stage tomato seedlings were exposed to 8 °C/5 °C for 2 days and sampled every day. All values shown are the mean ± SD (*n* = 3). Lowercase letters a–g indicate that mean values are significantly different among samples (*p* < 0.05).

**Figure 7 ijms-24-11042-f007:**
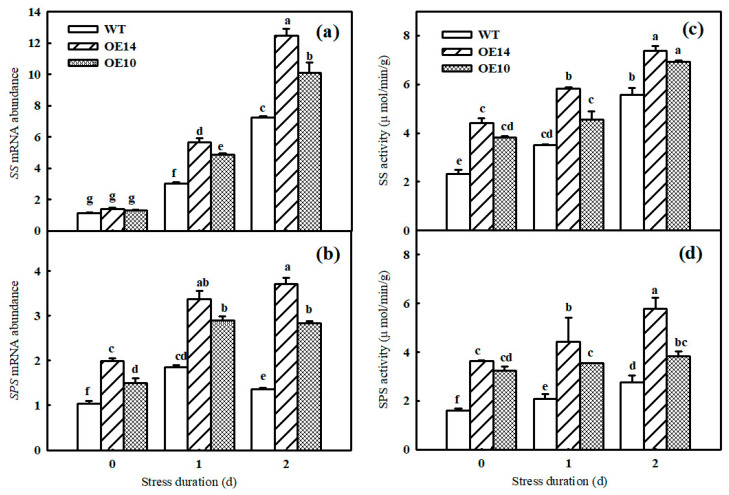
The effect of *SlTDC1* overexpression on the activities and relative mRNA expressions of carbon metabolism in tomato seedlings under chilling stress. (**a**,**b**) *SS* and *SPS* mRNA abundance. (**c**,**d**) Activities of sucrose synthase and sucrose phosphate synthase. The four-leaf-stage tomato seedlings were exposed to 8 °C/5 °C for 2 days and sampled every day. All values shown are the mean ± SD (*n* = 3). Lowercase letters a–g indicate that mean values are significantly different among samples (*p* < 0.05).

**Figure 8 ijms-24-11042-f008:**
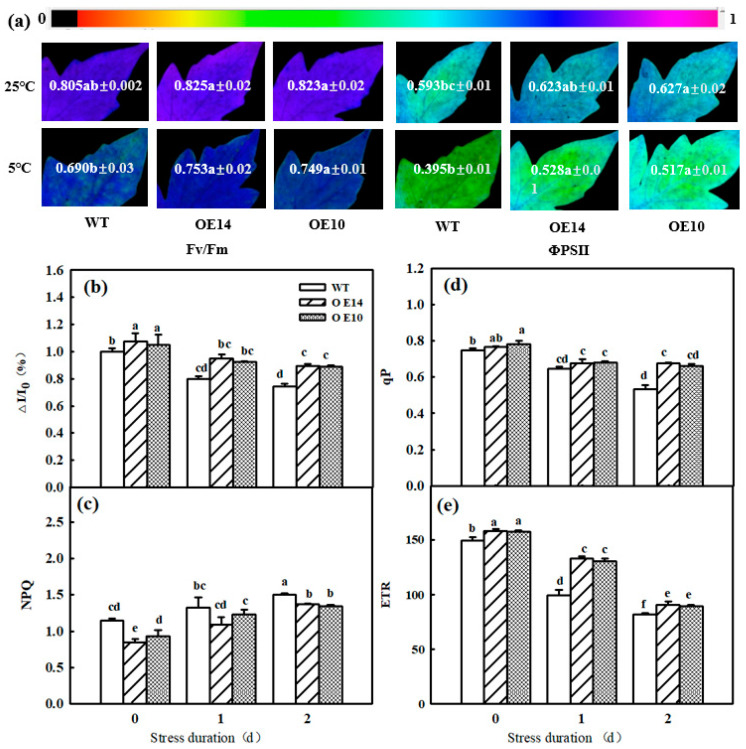
The effect of *SlTDC1* overexpression on the activities of PSII and PSI in tomato seedlings under chilling stress. (**a**) Images of Fv/Fm and ΦPSII, (**b**) ΔI/I_0_, (**c**) NPQ, (**d**) qP, and (**e**) ETR. The four-leaf-stage tomato seedlings were exposed to 8 °C/5 °C for 2 days and measured every day. All values shown are the mean ± SD (*n* = 3). Lowercase letters a–f indicate that mean values are significantly different among samples (*p* < 0.05).

**Figure 9 ijms-24-11042-f009:**
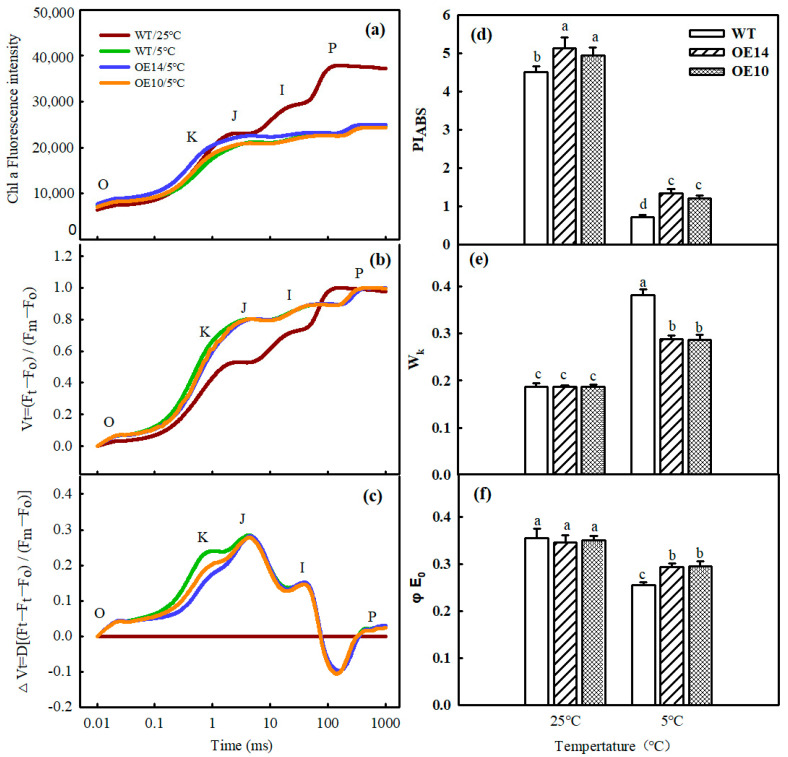
The effect of *SlTDC1* overexpression on the activities of PSII in tomato seedlings under chilling stress. FO: fluorescence at t = 0, FK: fluorescence at t = 300 μs, FJ: fluorescence at t = 2 ms, FI: fluorescence at t = 30 ms, FP: Maximum fluorescence. (**a**) Ch1a fluorescence intensity, (**b**) V_t_, (**c**) ΔV_t_, (**d**) W_k_, (**e**) PI_ABS_, and (**f**) φ_Eo_. The four-leaf-stage tomato seedlings were exposed to 8 °C/5 °C for 2 days and measured. All values shown are the mean ± SD (*n* = 3). Lowercase letters a–d indicate that mean values are significantly different among samples (*p* < 0.05).

**Figure 10 ijms-24-11042-f010:**
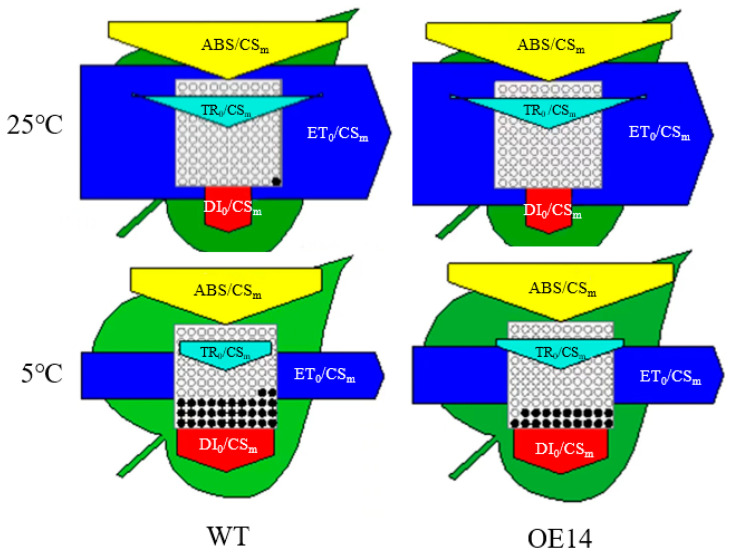
Energy pipeline leaf model of the phenomenon flux (per cross-section, CS) in the fourth fully expanded leaf of the tomato with *SlTDC1* overexpression and WT. Each relative value is drawn by the width of the corresponding arrow, representing a parameter. Hollow and full black circles indicate the percentage of active (QA reduction) and inactive (non-QA reduction) reaction centers of photosystem II (PSII), respectively; ABS/CS_m_, photon flux absorbed by pigment per CS antenna; TR/CS_m_, captured energy flux per CS; ET/CS_m_, electron transport flux per CS; D/CS_m_, non-photochemical quenching. The four-leaf-stage tomato seedlings were exposed to 8 °C/5 °C for 2 days and measured.

**Figure 11 ijms-24-11042-f011:**
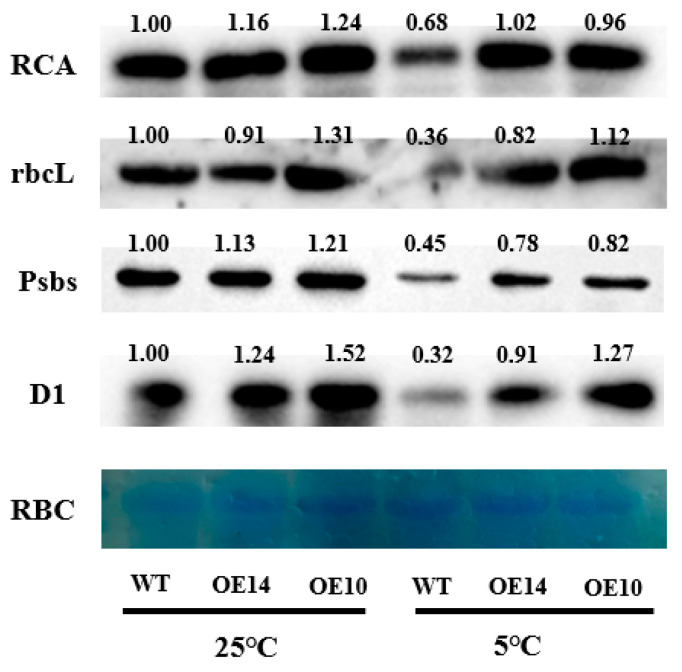
The effect of *SlTDC1* overexpression on the protein expression of RCA, rbcL, Psbs, and D1 of tomato seedlings under chilling stress. The four-leaf-stage tomato seedlings were exposed to 8 °C/5 °C for 2 days and sampled. The leaves for photosynthesis analysis were sampled for total protein extraction.

**Table 1 ijms-24-11042-t001:** Effects of *TDC1* overexpression and inhibition on sugar content under low-temperature stress. The four-leaf-stage tomato seedlings were exposed to 8 °C/5 °C for 2 days and sampled every day. All values shown are the mean ± SD (*n* = 3). Lowercase letters a–f indicate that mean values are significantly different among samples (*p* < 0.05).

Stress Days (d)	Strains	A_sat_ (μmol·m^−2^·s^−1^)	Total Sugar Content (mg·g^−1^DW)	Sucrose Content (mg·g^−1^DW)	Glucose Content (mg·g^−1^DW)	Fructose Content (mg·g^−1^DW)	Starch Content (mg·g^−1^DW)
0 d	WT	25.53 ± 1.28 b	234.30 ± 3.50 b	74.26 ± 1.35 b	6.61 ± 0.16 b	5.49 ± 0.05 c	128.54 ± 0.68 b
	*OE14*	28.3 ± 0.98 a	243.00 ± 2.14 a	84.21 ± 2.31 a	8.69 ± 0.40 a	7.10 ± 0.20 a	116.93 ± 2.05 c
	*OE10*	27.83 ± 0.6 a	244.56 ± 4.12 a	82.78 ± 3.96 a	8.92 ± 0.21 a	6.20 ± 0.15 b	117.58 ± 2.02 c
1 d	WT	10.73 ± 0.55 c	269.56 ± 1.56 b	115.11 ± 2.47 c	16.58 ± 0.49 b	8.49 ± 0.13 c	164.51 ± 2.50 d
	*OE14*	18.8 ± 0.45 a	353.27 ± 1.23 a	135.40 ± 1.26 a	19.25 ± 1.01 a	10.74 ± 0.03 a	142.98 ± 1.93 f
	*OE10*	17.26 ± 0.68 b	348.97 ± 6.95 a	129.55 ± 1.24 b	19.49 ± 0.17 a	9.87 ± 0.16 b	149.27 ± 0.79 e
2 d	WT	5.83 ± 0.6 b	324.40 ± 4.61 b	133.87 ± 4.67 b	31.04 ± 0.32 c	11.53 ± 0.07 c	215.72 ± 0.11 c
	*OE14*	10.26 ± 0.45 a	393.76 ± 5.09 a	150.97 ± 1.28 a	36.47 ± 0.47 a	17.54 ± 0.05 a	198.22 ± 2.73 d
	*OE10*	10.43 ± 0.68 a	393.60 ± 4.25 a	149.35 ± 3.56 a	33.91 ± 0.66 b	15.56 ± 0.13 b	202.66 ± 0.34 d

**Table 2 ijms-24-11042-t002:** Primers used the in quantitative RT-PCR.

Genes	Accession Numbers	Primer Pairs (5′-3′)
Actin	XM_001321306	TTTGCTGGTGATGATGCC
		CCTTAGGGTTGAGAGGTGCTT
TDC	XM_004243253	GCCTAAGCCCTCATAAGTGG
		TGCCAGTCCTTGTAATCCA
SS	XM_019211274	CGCCAAGAATCCACGACTAA
		TCTGCCTGCTCTTCCAAATC
SPS	XM_0012469991	CAGTCAGCAGAGAGGAAAGAAG
		CAGTATCAGAATCCCGTCCAAG
RCA	XM_010327541	GAAGAAACAGACTGATGGGGAC
		CGTTGGAGCCTGGAAAAGC
RbcL	XM_012015910	TTTCCAAGGTCCGCCTCA
		CCACCGCGAAGACATTCATA
RbcS	XM_026029055	CCATTGCTAGCAACGGTGGAAGA
		TGCTCGTCGGACAAATCAGG

## Data Availability

All data generated or analyzed during this study are included in this published article.

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
