# Peer review of "SlTDC1* Overexpression Promoted Photosynthesis in Tomato under Chilling Stress by Improving CO_2_ Assimilation and Alleviating Photoinhibition"

_ijms, 2023, doi:10.3390/ijms241311042_

Round 1

Reviewer 1 Report

Review IJMS 2456144

`Melatonin (MT), as a plant hormone, participates in the regulation of abiotic stress tolerance, including chilling stress. The manuscript by Liu and colleagues describes the effect of the overexpression of tomato Tryptophan decarboxylase gene 1 (SlTDC1) encoding the first rate-limiting gene for melatonin (MT) biosynthesis on the photosynthesis process under low temperatures. The authors convincingly prove that overexpression of SlTDC1 notably increase chlorophyll content, maintain normal chloroplast structure and positively affect photosynthetic parameters related to electron transport and energy distribution. Transgenic lines overexpressing SlTDC1 also display higher photosynthetic carbon assimilation and increased photosynthesis-related protein levels, thus alleviating the chilling damage to seedlings. The study makes a significant contribution to understanding the mechanisms of action of melatonin under chilling stress.

I have only some minor comments.

1. The statement that “chilling stress significantly promoted the accumulation of starch” (Lines 272-274) contradicts generally accepted view on the effect of low temperatures on carbohydrate metabolism. Obviously, this means slower synthesis of sugars and an increased starch content in WT as compared to overexpressing lines.

2. I would recommend moving the SFig.2 (The mRNA abundance (A) and endogenous melatonin content (B)) to the main text and add a protocol to “Materials and Methods” for determining melatonin

3. Fig 5 Images of chloroplast structure under normal temperature with the descriptions should be provided to allow assessment of the effects of chilling stress and overexpression of SlTDC1 per se.

3. The origin of the antibodies used for Westerns must be specified.

4. I would also recommend editing by a native speaker.

 In particular, the phrase “the tissues were sampled or measured for parameters at 0 d, 1 d and 2 d after chilling stress” (lines 117-118) is misleading, as it can be understood that the material was fixed 1 or 2 days after the end of the stress.

“Stress times” on captions to all figures should be changed for “stress duration”.

Fig, 4 line 248. “Photosynthetic enzyme” should be changed for “enzymes”

Editing  by a native speaker is highly desirable

Reviewer 2 Report

Comments:

Line18: Since the authors do not use abbreviation ‘EL’ and ‘H2O2’ in the Abstract, they should be removed;

Line 23-24: The same with Wk , φEo, and PIABS;

Throughout the text: References do not follow journal rules;

Line 53-57, Line 83-87: This sentence should be divided into two or more;

Line 71-72: The authors do not use (T5H) and ‘COMT’ in the text below. They should be deleted;

Line 76-78: Please, correct this sentence with especial attention to EL;

Line 80-82: The correctness of the wording in this sentence needs to be checked;

Linу 190: The abbreviation ‘WT’ has already been used in the text above. Don’t repeat it here;

Line 196: The same with ‘EL’;

Line 188-195: SFig 2 should be placed in the main text;

Line 187: It seems better to use " Electrolyte leakage rate and H2O2 content’ instead of ‘Overexpression of SlTDC1 improved chilling tolerance " as the title of this subsection;

Line 226: In the section MandM describe, please, what Gs,  E, and Ci mean, and how they were measured;

Line 213: According to Fig2, the increase is about 40% only for the Pn of OE14 and OE10 seedlings, no? Check this, please;

Line 217: What Tr means? The same as ‘E’? Correct, please;

Line 234: It seems better to use " Gene expressin and photosynthetic enzyme activity…’ instead of ‘Overexpression of SlTDC1 promoted gene expression and photosynthetic enzyme activity under chilling stress " as the title of this subsection;  

Line 271: It seems better to use " chlorophyll content and chloroplast ultrastructure ..’ instead of ‘Overexpression of SlTDC1 increased chlorophyll content and maintained chloroplast ultrastructure under chilling stress " as the title of this subsection;  

The same with other titles of subsections;

 Line 271-283: Don't generalize sugar and starch content to carbon metabolism;

Fig7a. In my opinion, Fv/Fm and ΦPSII data are shown unsuccessfully;

Line 152: What Wk, PIABS, φEo,ABS/CSm, TR/CSm, D/CSm and E/CSm mean?

Line 322: According to Fig8d 89/12% is not true;

Line 329-337: this section is unclear because it is not known what Wk, PIABS, φEo,ABS/CSm, TR/CSm, D/CSm and E/CSm mean;

No link to figure 9;

Line 368-369: This sentence needs Referenses;

Line 408: If ‘ a decline in Fv/Fm and ΔI/I0, which were higher in SlTDC1-overexpress- 408 ing tomato seedlings than in WT seedlings’ how ‘ the overexpression of SlTDC1 alleviated the photoinhibition of PSII and PSI’?

Line 435: Use ‘WT plants’ instead of ‘WT’. The same with ‘SlTDC1 overexpression’;

Line 439: Don’t start the sentences with abbrewiation.

Round 2

Reviewer 1 Report

Review 2 IJMS 2456144

After the changes and additions made, the manuscript “SlTDC1 overexpression promoted photosynthesis in tomato under chilling stress by improving CO2 assimilation and alleviating photoinhibition” can be submitted for publication. I would only recommend adding "melatonin synthesis" to the keywords, as the authors attribute better tomato cold tolerance to melatonin content. This will undoubtedly make the manuscript attractive to a wider range of researchers.

Minor concerns:

Line 82 “affected” to “affecting”

Line 441 “has” to “have”

 Minor editing of English language  is desirable

Reviewer 2 Report

The manuscript was corrected according the comments